# Research on the Analytical Model of Improved Magnetic Flux Leakage Signal for the Local Stress Concentration Zone of Pipelines

**DOI:** 10.3390/s22031128

**Published:** 2022-02-02

**Authors:** Lijian Yang, Fuyin Zheng, Ping Huang, Shi Bai, Yuming Su

**Affiliations:** College of Information Science and Engineering, Shenyang University of Technology, Shenyang 110870, China; yanglijian888@163.com (L.Y.); zhengfuyin666@163.com (F.Z.); huangping0809@sut.edu.cn (P.H.); suhei315@163.com (Y.S.)

**Keywords:** local stress concentration, magnetomechanical model, analytical model, MFL signal

## Abstract

Local stress concentrations pose a significant hazard to the safe operation of pipelines. However, the classical analytical model of the magnetic flux leakage (MFL) signal is still unable to effectively quantitatively analyze and accurately evaluate the local stress concentration zone of a pipeline. In this paper, based on the Jiles–Atherton model of the magnetomechanical effect, the mathematical relationship between stress and the magnetization of ferromagnetic material under hysteresis conditions is introduced, and an improved analytical model of the MFL signal based on the magnetomechanical model is established. The influence law of stress intensity on the MFL signal in the local stress concentration zone of the pipeline is calculated and analyzed, and the theoretical calculation results are verified through experiments. Simulation and experimental results show that, considering the hysteresis condition, the stress causes a change in the hysteresis loop of the ferromagnetic material, and the magnetization strength of the material decreases with increasing stress; the effect of stress on the magnetization strength of ferromagnetic materials is most obvious when the external magnetic field is approximately 5 KA/m. The MFL signal on the surface of the local stress concentration zone of the pipe changes abruptly, and the amount of change in the axial amplitude and radial peak-to-peak value of the leakage signal of the pipe tends to increase with the increase in the stress intensity of the local stress concentration zone. A comparison of the analysis with the classical analytical model of the MFL signal shows that the improved analytical model of the MFL signal is more suitable for the quantification study of the local stress concentration zone of the pipeline.

## 1. Introduction

Long-distance oil and gas pipelines play an important role in the transportation of oil, gas, and other energy sources, which is why ensuring the safe operation of pipelines is of great economic and social importance [1]. In-service pipelines are subjected to external loads, high temperature, and high pressure for a long time. Micro-damage will occur in the local zone of the pipe wall, resulting in local stress concentration and threatening the safe operation of the pipeline [2]. The external load is one of the reasons for the local stress concentration of the pipeline. The buried pipeline is sensitive to the movement of the soil. When soil subsidence, landslide, and seismic subsidence occur, the pipeline will produce local stress concentration under the action of external load, so that the pipeline is in a dangerous state of local instability, and serious distortion deformation, fracturing, and fracture may occur. Conventional non-destructive testing technologies, such as magnetic powder, magnetic flux leakage (MFL), electromagnetic, and ultrasonic [3,4,5,6], have played an important role in defect detection and accident prevention in pipelines. However, they can only find macroscopic volume defects that have formed and cannot effectively evaluate the local stress concentration of the pipeline, so that sudden accidents caused by local stress damage cannot be avoided [7].

Currently, MFL is an effective non-destructive inspection method for pipelines, with the advantages of high interference immunity, fast signal acquisition, and no coupling agent. In-depth research on theoretical calculations, magnetization intensity, detection methods, signal processing, and quantitative characterization of magnetic leakage detection has been conducted by international scholars. Battelle developed a dual-field MFL detector [8]. Experimental research verified that the MFL signal is mainly affected by the change of pipe wall geometry at a high magnetic field level. At a low magnetic field level, the MFL signal is caused by pipe wall local stress and material metallurgical changes. Conventional MFL detection is mostly based on oversaturation magnetization to make the detection signal clearer. Its mathematical model can only analyze the dimensional characteristics of a single defect and cannot effectively characterize other signals such as pipe local stress. In actual engineering, the stress of the pipeline at the external load is much greater than the average value of the stress of the pipeline, which will form a local stress concentration and lead to deformation and fracture of the pipeline in serious cases. However, it is difficult to evaluate the pipeline local stress using the traditional mathematical model. To clarify the mathematical relationship between local stress and magnetic signal, scholars have conducted in-depth research. According to Jiles and Atherton, the magnetization of magnetic materials caused by the applied cyclic stress is mainly affected by the irreversible change of magnetization toward the hysteresis-free magnetization curve [9,10]. Liu calculated and analyzed the distribution pattern of magnetic field leakage on the pipe surface and the trend of change with stress by building a magnetomechanical model for pipes under complex stress conditions [11,12,13]. Two analytical models based on the magnetic dipole analytical model and a finite element numerical model based on the Maxwell equation are mainly used to address the problem of analytical calculation of magnetic field leakage. Zatsepin et al. proposed a magnetic dipole model that uses magnetic dipoles (dots, lines, and bands) to model defects in conductive samples [14]. By improving the magnetic dipole model, Wang effectively explained the phenomenon where the normal component of the magnetic field leakage crosses the zero point and the tangential component appears to have extreme value, and analyzed the relationship between magnetic field leakage distribution and stress [15]. With the development of computers, scholars in various countries have used the finite element method to study the solution of magnetic field leakage caused by pipeline stress and optimized the magnetization method of ferromagnetic materials [16,17,18,19]. Zhang created a detection system based on the magnetoelastic effect theory, which uses the method of electromagnetic transmission to detect and analyze steel stress [20]. Pitman calculated the hysteresis loop of ferromagnetic material under tensile and compressive stress, and the stress influence law on ferromagnetic material’s magnetic properties was described and assessed [21]. Luo updated the J–A model by combining the Jiles–Atherton (J–A) model with the Zheng–Jing–Liu Xing–En (Z–L) model, which improved the consistency of the magnetization curve and hysteresis loop of ferromagnetic materials and provided a theoretical framework for quantitative stress analysis [22,23]. Wu simulated the stress distribution of a natural gas submarine pipeline in a complex and changeable regional environment, which provided a reliable reference for the stress parameters in pipeline operation and a theoretical basis for pipeline safety construction [24].

In this paper, the distribution characteristics of the MFL signal in the local stress concentration zone of the pipeline under external load are analyzed in depth. The improved analytical model of the MFL signal based on the magnetomechanical model is established. The influence of stress intensity on the hysteresis loop of Q235 pipeline steel and the variation law of the MFL signal in the local stress concentration zone of pipeline with stress intensity is calculated and analyzed. Theoretical simulation and experimental results provide a scientific basis for the detection and quantitative study of the local stress concentration zone in the pipeline.

## 2. Theoretical Model

The MFL signal in the local stress concentration zone of the pipeline is generated by the superposition of the external magnetic field and the local stress. In this paper, the oil and gas pipeline (length of 6000 mm, diameter of 273 mm, wall thickness of 7 mm; the material is Q235 steel) is the research object. The influence of stress intensity on the MFL signal in the local stress concentration zone of the pipeline is mainly discussed. The high magnetic permeability of ferromagnetic materials is the basis for MFL technology. Using a steel brush, the excitation unit magnetizes the entire wall. The magnetic line is parallel to the pipeline wall when there is no damage; however, when damage exists on the pipe, the magnetic lines are hindered by the damaged zone, causing part of the magnetic lines to leak onto the pipe surface and create an MFL signal. The magneto sensors detect the MFL signal to assess whether or not there is pipeline damage. The excitation unit, steel brush, yoke iron, sensors, odometer wheels, and data storage system are common components of MFL detection systems. A three-dimensional diagram of the pipeline and MFL detection system is shown in Figure 1.

As shown in Figure 1, the axial direction of the pipeline is the *x*-axis, the circumferential direction is the *y*-axis, and the radial direction is the *z*-axis. During MFL detection, the detection system advances along the axial direction. The detection probe is located at the center of the two magnetic poles with slight lift-off.

### 2.1. Classical Analytical Model of MFL Signal

According to the Biot–Savart law, the basic formula of the classical analytical model of MFL signal for pipeline defects is [25]
(1)$$dH=\frac{\rho_{s}dS}{4\pi \mu_{0}{\left|r_{0}\right|}^{3}}\vec{r_{0}},$$
where *H* is the magnetic field strength of the space field point. According to the definition of magnetic charge density [26], *ρ_s_* = *μ*_0_Δ*M* is the magnetic charge density of the defect sidewall (Δ*M* is the magnetization difference between the two sides of the magnetic charge surface), *dS* is the area element of the magnetic charge surface, *μ*_0_ is the vacuum permeability, *μ*_0_ = 4*π* × 10^−7^ H/m, $\vec{r_{0}}$ is the direction vector of the magnetic charge surface to space field point, and *r*_0_ is the distance from the magnetic charge surface to field point.

To facilitate the analysis, the iron pipe wall can be assumed to be subjected to the combined action of magnetic field and local stress. Similar to the MFL model at the defect, a magnetic charge is accumulated at the end of the pipe local stress concentration zone in the magnetization direction, and a scattering magnetic field is generated. With the surface center of the pipeline local stress concentration zone taken as the origin, the classical model of the three-dimensional pipeline local stress concentration zone is established, as shown in Figure 2. The applied magnetic field *H*_0_ is parallel to the *x*-axis direction, the stress damage occurs in the local stress concentration zone, and the damage is uniform. All the insurmountable pinning points occur in the local stress concentration zone. The magnetization intensity in the local stress concentration zone is *M_in_*, and the magnetization intensity outside the zone is *M_out_*.

The locally varying magnetic field in the local stress concentration zone is equated with a uniformly distributed magnetic charge on the end face of the area, and the magnetic charge surface density is *ρ_s_*, −*ρ_s_*. The three-dimensional space field point coordinates are defined as *P* (*x*, *y*, *z*), and the magnetic charge surface element coordinates are defined as (*x_m_*, *y_m_*, *z_m_*). Then, the magnetic field intensity generated by the surface element at the space point *P* is
(2)$$dH=\frac{\rho_{s}dy_{m}dz_{m}}{2\pi \mu_{0}{\left|r\right|}^{3}},$$
where *r* is the vector distance from the magnetic charge surface to the field point. As shown in Figure 2, the wall region is ±*x_m_*, 0 to −*D_z_*, −*D_y_* to *D_y_*. The local magnetic field *H* is the synthetic magnetic field generated by the combined action of positive and negative magnetic charges on the left and right ends of the local stress concentration zone. Thus, the following integration is performed on *dH*, and the axial (*x*-direction) and radial (*z*-direction) components of *H* are obtained as follows:(3)$$\begin{array}{cc} H_{x} & =\frac{\left(M_{out}-M_{in}\right)dy_{m}dz_{m}}{4\pi }\times [\int_{-D_{y}}^{D_{y}}\int_{-D_{z}}^{0}\frac{x+x_{m}}{{\left[{\left(x+x_{m}\right)}^{2}+{\left(y-y_{m}\right)}^{2}+{\left(z-z_{m}\right)}^{2}\right]}^{2}}\\ & -\int_{-D_{y}}^{D_{y}}\int_{-D_{z}}^{0}\frac{x-x_{m}}{{\left[{\left(x-x_{m}\right)}^{2}+{\left(y-y_{m}\right)}^{2}+{\left(z-z_{m}\right)}^{2}\right]}^{2}}] \end{array}$$
(4)$$\begin{array}{cc} H_{z} & =\frac{\left(M_{out}-M_{in}\right)dy_{m}dz_{m}}{4\pi }\times [\int_{-D_{y}}^{D_{y}}\int_{-D_{z}}^{0}\frac{z-z_{m}}{{\left[{\left(x+x_{m}\right)}^{2}+{\left(y-y_{m}\right)}^{2}+{\left(z-z_{m}\right)}^{2}\right]}^{2}}\\ & -\int_{-D_{y}}^{D_{y}}\int_{-D_{z}}^{0}\frac{z-z_{m}}{{\left[{\left(x-x_{m}\right)}^{2}+{\left(y-y_{m}\right)}^{2}+{\left(z-z_{m}\right)}^{2}\right]}^{2}}] \end{array}$$

### 2.2. Improved Analytical Model of MFL Signal

Local stress concentrations occur in the local zone of the pipe under external loads, and local stress is considered to cause a change in the magnetic properties of the pipe material, which in turn influences the leakage signal. Therefore, local stress and other parameters need to be introduced into the classical analytical model of the MFL signal when performing the analytical calculation of the MFL signal in the local stress concentration region of the pipe. In the equilibrium state of the magnetomechanical effect, based on the Langevin function, the equation of state of the magnetization of the pipe in the non-hysteresis state can be expressed as [27]
(5)$$M_{an}=M_{s}\left[\coth\left(\frac{H_{e}}{a}\right)-\frac{a}{H_{e}}\right],$$
where *M_an_* is the non-hysteresis magnetization, A/m; *M_s_* is the saturation magnetization, A/m; *H_e_* is the effective magnetic field intensity, A/m; *a* is the magnetization model parameter, A/m; and *H_e_* is the effective magnetic field strength under the combined action of external magnetic field and stress. According to the magnetostrictive expression in the literature [28], *H_e_* can be expressed as
(6)$$H_{e}=H+\alpha M-N_{d}M+\frac{3\sigma }{\mu_{0}}\left(iM_{}^{2i-1}\sum_{n=0}^{\infty }\frac{\sigma^{n}}{n!}\gamma_{i}^{n}\left(0\right)\right),$$
where *H* is the external magnetic field intensity, A/m; *σ* is the stress, MPa; *α* is the internal coupling field coefficient; *M* is the magnetization of the material, A/m; *N_d_* is the demagnetization coefficient; and *γ_i_^n^*(0) is the n-order derivative of the material correlation coefficient *γ_i_* corresponding to *σ* = 0, generally *i* = *n* = 1.

Under the action of the external magnetic field, the magnetization *M* of ferromagnetic material consists of the reversible magnetization *M_rev_* of domain wall bending and the irreversible magnetization *M_irr_* of domain wall displacement [29], and *M* = *M_rev_* + *M_irr_*, *M_rev_* = *c*(*M_an_* − *M_irr_*), *c* is the reversible coefficient. The differential equation of the irreversible magnetization *M_irr_* with respect to the effective magnetic field *H_e_* can be expressed as
(7)$$\frac{dM_{irr}}{dH_{e}}=\frac{\mu_{0}}{k_{eff}\delta }\left(M_{an}-M\right),$$
where *k_eff_* is the effective pinning coefficient, and δ is the directional coefficient. When *dH_e_*/*dt* > 0, δ = 1; otherwise, δ = −1. The simultaneous Equations (5)–(7) are obtained and the differential equations of the external magnetic field *H* of the magnetization *M* are expressed as follows:(8)$$\begin{array}{cc} \frac{dM}{dH} & =\left[\frac{\mu_{0}}{k_{eff}\delta }\left(M_{an}-M\right)+\frac{c}{c+1}\frac{dM_{an}}{dH}\right]\\ & \times {\left\{\frac{1}{1-c}-\frac{\mu_{0}}{k_{eff}\delta }\left(M_{an}-M\right)\left[\alpha -N_{d}+\frac{3\sigma }{\mu_{0}}\left(\gamma_{1}\left(0\right)+{\gamma^{\prime }}_{1}\left(0\right)\sigma \right)\right]\right\}}^{-1} \end{array}$$

Among them, *dM_an_*/*dH* can be obtained by simultaneous Equations (5) and (6), i.e.,
(9)$$\begin{array}{cc} \frac{dM_{an}}{dH} & =-M_{s}\left[\frac{1}{a}{\mathrm{csch}}^{2}\left(\frac{H_{e}}{a}\right)-\frac{a}{H_{e}^{2}}\right]\\ & \times {\left\{1+M_{s}\left[\frac{1}{a}{\mathrm{csch}}^{2}\left(\frac{H_{e}}{a}\right)-\frac{a}{H_{e}^{2}}\right]\left[\alpha -N_{d}+\frac{3\sigma }{\mu_{0}}\left(\gamma_{1}\left(0\right)+{\gamma^{\prime }}_{1}\left(0\right)\sigma \right)\right]\right\}}^{-1} \end{array}$$

The variation of magnetization *M* with external magnetic field *H* under different stresses can be calculated by Formulas (8) and (9).

In the actual pipeline stress detection, the stress distribution in the pipeline local stress concentration zone is a function of continuous change. This continuous change causes the magnetic charge to accumulate inside the pipeline local stress concentration zone, and the volume magnetic charge density [30] can be expressed as
(10)$$\begin{array}{cc} \rho_{v} & =-\mu_{0}\nabla M\\ & =-\mu_{0}\left(\frac{\partial M_{x}}{\partial x}+\frac{\partial M_{y}}{\partial y}+\frac{\partial M_{z}}{\partial z}\right) \end{array}$$

Equation (10) is a stress magnetic charge model. Therefore, the magnetic field intensity at any point in space can be expressed as
(11)$$H=\int_{-z}^{0}\int_{y_{1}}^{y_{2}}\int_{x_{1}}^{x_{2}}\frac{\rho_{v}dV}{4\pi \mu_{0}{\left|r\right|}^{3}}\vec{r},$$
where *z* is the depth of the local stress concentration zone, *y*_1_ and *y*_2_ are the circumferential intervals of the local stress concentration zone, *x*_1_ and *x*_2_ are the axial intervals of the local stress concentration zone, *dV* is the volume element of the volume magnetic charge, $\vec{r}$ is the vector from the volume magnetic charge element to the spatial field point, and *r* is the distance from the volume magnetic charge element to the field point.

With the center of the upper surface of the local stress concentration zone (located in the center of the magnetization zone) of the pipeline taken as the origin, the magnetic field *H*_0_ of the magnetization zone is parallel to the *x*-axis (axial) direction, the direction perpendicular to the pipeline surface is the *z*-axis (radial), and the direction perpendicular to the external magnetic field of the upper surface of the local stress concentration zone is the *y*-axis (circumferential). A three-dimensional rectangular coordinate system is established, as shown in Figure 3.

In Figure 3, the size of the local stress concentration zone is 2*D_x_*_1_ in length, 2*D_y_*_1_ in width, and *h* in depth. With the assumption that the stress intensity distribution in the local stress concentration zone of the pipeline is linearly distributed with the center of the local stress concentration zone as the symmetry axis, the stress intensity at the center of the local stress concentration zone of the pipeline is set to be the largest, the size is *σ*_max_, and the edge stress of the local stress concentration zone is *σ*_0_.

The normal stress distribution function *σ*(*x*) along the axial direction in the local stress concentration zone can be obtained as follows:(12)$$\left\{ \begin{array}{l} \sigma \left(x\right)=\frac{\sigma_{\max}-\sigma_{0}}{D_{x1}}x+\sigma_{\max}\\ \sigma \left(x\right)=-\frac{\sigma_{\max}-\sigma_{0}}{D_{x1}}x+\sigma_{\max} \end{array}\right.\begin{array}{l} -D_{x1}\leq x\leq 0\\ 0<x\leq D_{x1} \end{array}$$

According to the material mechanics [31], the strain energy *W* per unit volume of the material under unidirectional stress can be expressed as
(13)$$W=\frac{\sigma^{2}}{2E},$$

In the formula, *E* is the Young’s modulus, GPa. Therefore, the differential of magnetization intensity to strain energy can be expressed as
(14)$$\begin{array}{cc} \frac{dM}{dW} & =\frac{dM_{rev}}{dW}+\frac{dW_{irr}}{dW}\\ & =\frac{1-c}{\xi }\left(M_{an}-M_{irr}\right)+c\frac{dM_{an}}{dW} \end{array}$$

In the formula, *ξ* is a coefficient, which is related to the energy per unit volume. *dW* can be deduced from Formula (13) to *dW* = (*σ*/*E*)*dσ*, and the differential equation of stress magnetization under stress state is Formula (15)
(15)$$\begin{array}{cc} \frac{dM}{d\sigma } & =\frac{\sigma }{E\xi }\left(1-c\right)\left(M_{an}-M_{irr}\right)+c\frac{dM_{an}}{d\sigma }\\ & =\frac{\sigma }{E\xi }\left(M_{an}-M\right)+c\frac{dM_{an}}{d\sigma } \end{array}$$

Combining Equations (10), (12), and (15), the distribution function of the bulk magnetic charge density ([0, *D_x_*_1_] magnetic charge density is opposite) along the *x*-axis [−*D_x_*_1_, 0] pipeline local stress concentration zone can be expressed by Equation (16)
(16)$$\begin{array}{cc} \rho_{v} & =-\mu_{0}\frac{\partial M_{x}}{\partial x}=-\mu_{0}\frac{\partial M_{x}}{\partial \sigma \left(x\right)}\frac{\partial \sigma \left(x\right)}{\partial x}\\ & =-\frac{\mu_{0}\left(\sigma_{\max}-\sigma_{0}\right)}{D_{x1}}\times \langle \frac{\sigma \left(x\right)}{E\xi }\left(M_{an}-M\right)-cM_{s}\left[\frac{1}{a}{\mathrm{csch}}^{2}\left(\frac{H_{e}}{a}\right)-\frac{a}{H_{e}^{2}}\right]\\ & \times \left\{\frac{3}{\mu_{0}}\left[\left(\gamma_{11}+\gamma_{12}\sigma \left(x\right)\right)M+2\left(\gamma_{21}+\gamma_{22}\sigma \left(x\right)\right)M^{3}\right]\right\}\rangle \\ & \times \langle 1+cM_{s}\left[\frac{1}{a}{\mathrm{csch}}^{2}\left(\frac{H_{e}}{a}\right)-\frac{a}{H_{e}^{2}}\right]\\ & \times {\left\{\alpha -N_{d}+\frac{3\sigma \left(x\right)}{\mu_{0}}\left[\left(\gamma_{11}+\gamma_{12}\sigma \left(x\right)\right)+6\left(\gamma_{21}+\gamma_{22}\sigma \left(x\right)\right)M^{2}\right]\right\}\rangle }^{-1} \end{array}$$
where *γ11*, *γ12*, *γ21*, and *γ22* are the material correlation coefficients. According to the improved analytical model of the MFL signal, a large number of volume magnetic charges accumulate in the local stress concentration zone of the pipeline. Combining (5), (12), and (16), the expressions of the axial and radial components of the leakage magnetic field of the surface field point *P*(*xi*, *yi*, *zi*) in the local stress concentration zone of the pipeline during the MFL detection process are
(17)$$\begin{array}{cc} H_{x} & =\frac{1}{4\pi \mu_{0}}[\int_{-h}^{0}\int_{-D_{y1}}^{D_{y1}}\int_{-D_{x1}}^{0}\frac{\rho_{v}\left(xi-x\right)dxdydz}{\left[{\left(x-xi\right)}^{2}+{\left(y-yi\right)}^{2}+{\left(z-zi\right)}^{2}\right]}\\ & -\int_{-h}^{0}\int_{-D_{y1}}^{D_{y1}}\int_{0}^{D_{x1}}\frac{\rho_{v}\left(xi-x\right)dxdydz}{\left[{\left(x-xi\right)}^{2}+{\left(y-yi\right)}^{2}+{\left(z-zi\right)}^{2}\right]}] \end{array}$$
(18)$$\begin{array}{cc} H_{z} & =\frac{1}{4\pi \mu_{0}}[\int_{-h}^{0}\int_{-D_{y1}}^{D_{y1}}\int_{-D_{x1}}^{0}\frac{\rho_{v}\left(zi-z\right)dxdydz}{\left[{\left(x-xi\right)}^{2}+{\left(y-yi\right)}^{2}+{\left(z-zi\right)}^{2}\right]}\\ & -\int_{-h}^{0}\int_{-D_{y1}}^{D_{y1}}\int_{0}^{D_{x1}}\frac{\rho_{v}\left(zi-z\right)dxdydz}{\left[{\left(x-xi\right)}^{2}+{\left(y-yi\right)}^{2}+{\left(z-zi\right)}^{2}\right]}] \end{array}$$

Equations (17) and (18) can be used to calculate and analyze the leakage magnetic field in the local stress concentration zone of the pipeline to realize the detection and quantitative analysis of the local stress concentration of the pipeline.

## 3. Analytical Calculation and Analysis

The local stress concentration of the pipeline was numerically calculated using the improved analytical model of the MFL signal. The local stress concentration zone dimension of the pipe is assumed to be 20 mm in length, 10 mm in width, and 7 mm in depth. When the external magnetic field is 5 KA/m, the maximum stress intensity in the local stress concentration zone of the pipeline is 200 MPa, and the average stress of the pipeline is about 0 MPa. The center of the local stress concentration zone is the origin. With the center of the local stress concentration zone taken as the origin, when the scanning path is from −30 mm to 30 mm along the *x*-axis and 1 mm away from the surface of the pipe wall, according to Formulas (17) and (18), the axial and radial component of the leakage magnetic field in the local stress concentration zone of the pipeline under the scanning path can be obtained as shown in Figure 4.

The model parameters in Figure 4 are *E* = 207 GPa, *ξ* = 24.5 × 10^3^ Pa, *α* = 0.00144, *N_d_* = 5 × 10^−5^, *Ms* = 1.56 × 10^6^ A/m, *a* = 900 A/m, *c* = 0.15, *γ*_11_ = 7 × 10^−24^A^−2^·m^2^, *γ*_12_ = −1 × 10^−25^A^−2^·m^2^·Pa^−1^, *γ*_21_ = −3.3 × 10^−30^A^−4^·m^4^, *γ*_22_ = 2.1 × 10^−38^A^−4^·m^4^·Pa^−1^. As can be seen from Figure 4, the MFL signal on the surface of the local stress concentration in the pipe is abnormal. The axial component of the leakage magnetic field has an extreme point above the centerline of the local stress concentration zone and is axially symmetrically distributed. The absolute value of the difference between the maximum and minimum values of the axial component is called the axial amplitude. The radial component of the leakage magnetic field is centrosymmetrically distributed, and two extreme points are found near the edge of the local stress concentration zone. The maximum and minimum points of the radial component are called radial peak-to-peak value (Vpp).

### 3.1. Effect of Stress on the Hysteresis Loop of Ferromagnetic Materials

When the stress intensity of ferromagnetic material is 50, 100, 150, and 200 MPa, the model parameters are consistent with Figure 4, where *k_eff_* = 1.522 × 10^−3^ A/m. According to Formulas (8) and (9), the hysteresis loops of ferromagnetic materials under different stresses can be obtained as shown in Figure 5.

Figure 5 shows that with the increase in the stress intensity, the shape of the hysteresis loop of the pipeline changes, and the magnetization of the material tends to decrease. With the increase in the magnetic field intensity, the difference of the magnetization of the hysteresis loop under different stresses decreases, and the stress has the most significant effect on the magnetization of the material at about 5 KA/m. The results are consistent with the results of the Battle experiment and engineering application [8]. In practical production applications, due to uneven doping, the magnetic mechanical characteristic parameters of the pipeline materials are different, and the hysteresis loops of different materials are slightly offset [32].

### 3.2. Effect of Stress on Leakage Magnetic Field of Local Stress Concentration in Pipeline

The classical analytical model of the MFL signal and the improved analytical model of the MFL signal is used to calculate the MFL signal of the pipeline local stress concentration zone under different stress intensity. The external magnetic field intensity is set to 5 KA/m, the stress intensity in the normal zone of the pipeline is about 0 MPa, and the maximum stress intensity in the local stress concentration zone is 0, 50, 100, 150, and 200 MPa. When other parameters are consistent with Figure 3, according to Equations (3) and (4), the axial and radial components of pipeline local stress concentration leakage magnetic field under different stresses in the classical MFL model under the hysteresis condition are shown in Figure 6.

As shown in Figure 6, in the classical analytical model of the MFL signal, two maximum points and two minimum points are present in the axial component of the MFL signal, which are axially symmetrically distributed. The radial component is distributed symmetrically about the center of the local stress concentration zone, and peaks or troughs appear at the edge of the local stress concentration zone. When the maximum stress intensity of the pipeline local stress concentration zone is 0, 50, 100, 150, and 200 MPa, the variation of axial component amplitude and radial component peak-to-peak value of the leakage magnetic field in the pipeline local stress concentration zone under the classical analytical model of the MFL signal is extracted as shown in Table 1.

Table 1 shows that based on the classical analytical model of MFL signal, the axial component amplitude and radial component peak-to-peak value of the MFL signal in the local stress concentration zone of the pipeline increase with the increase in the stress intensity in the local stress concentration zone.

Under the condition of hysteresis, according to Formulas (17) and (18), the axial and radial component distribution of the leakage magnetic field in pipeline local stress concentration caused by different stress intensity based on the improved analytical model of the MFL signal can be obtained, as shown in Figure 7.

As can be seen from Figure 7, the axial component distribution of the MFL signal in the improved analytical model of MFL signal is slightly different from that in the classical model, with only one minimum value and with an axially symmetrical distribution on the minimum point, and the axial and radial components of the MFL signal change more slowly. Table 2 shows the variation of axial component amplitude and radial component peak-to-peak value of the leakage magnetic field in the local stress concentration zone of pipeline based on the improved analytical model of the MFL signal when the maximum stress intensity of the local stress concentration zone is 0, 50, 100, 150, and 200 MPa.

The comparison between Table 1 and Table 2 shows that the variation of the MFL signal in the local stress concentration zone of the pipeline based on the improved analytical model of MFL signal is the same as that of the classical analytical model of the MFL signal. The peak-to-peak values of the axial component amplitude and radial component of the MFL signal increase with the increase in stress intensity in the local stress concentration zone, and the variation of axial component and radial component is stronger than that of the classical model.

## 4. Experiments and Result Analysis

### 4.1. Effect of Stress on Hysteresis Loop

To verify the influence of stress on the magnetic parameters of pipeline materials, the magnetic characteristic test under different stress is designed. The experimental material is a Q235 steel strip with the length of 500 mm, width of 15 mm, and thickness of 12 mm to simulate the actual pipeline. The two ends of the steel bar are fixed on the clamp of the tension machine. At the same time, the excitation coil is uniformly wound on the U-type magnetic core (the magnetic core is Q235 material), and the U-type magnetic core bipolar boot is fixed at the center of the Q235 steel bar. The receiving coil is wound on the steel bar between the two pole shoes, and the excitation coil and the receiving coil are connected to the excitation and receiving ends of the magnetic characteristic analyzer. Experimental magnetization curves and hysteresis loops under different stresses are shown in Figure 8.

The excitation signal of the excitation coil and the detection signal of the receiving coil can be collected simultaneously by the magnetic characteristic analyzer. When the stress is 0, 80, and 160 MPa, the hysteresis loop of the material under different stress are obtained, as shown in Figure 9.

As can be seen from Figure 9, with the increase in magnetic field strength, the magnetization curve and hysteresis loop of Q235 material show an overall decreasing trend with the increase in stress. When the external magnetic field reaches 5 KA/m, the change of the magnetic induction intensity is most obvious. The magnetic induction intensity of the material decreases with the increase in stress. The experimental results are in good agreement with the theoretical derivation. due to the nature of the material, stress affects coefficients such as α and K. They have been set as fixed values for ease of calculation. As a result, when the stress reaches 160 MPa, the simulation results will tilt somewhat.

The stress has an evident influence on the hysteresis loop when the external magnetic field strength is 5 KA/m. We set the maximum external magnetic field strength to 5 KA/m and the hysteresis loops of the experimental material at 0, 100 and 200 MPa were tested. Figure 10 depicts the results.

As indicated in Figure 10, the variation law is compatible with Figure 9. The magnetic induction intensity of the experimental material diminishes with increasing stress intensity when the external magnetic field exceeds 5 KA/m.

### 4.2. Distribution Characteristics of MFL Signal in the Local Stress Concentration of Pipeline

To verify the correctness and effectiveness of the improved analytical model of the MFL signal in the local stress concentration zone of the pipeline, a triaxial magnetic mechanics experiment was designed. In this paper, a single Q235 pipe is selected as the experimental material. The two ends of the pipe are fixed, and the radial external load is applied at the center below the pipe. The MFL internal detector of the pipe is dragged to make it pass through the pipe uniformly, and the three-axis data of the leakage magnetic field of the pipe are collected.

#### 4.2.1. Experimental Method

The experimental material is the Q235 pipeline with a length of 6000 mm, a diameter of 273 mm, and a wall thickness of 7 mm. Both ends of the pipeline are fixed with iron frames. The central position below the pipeline is used to simulate the external load using a jack to apply external force. To avoid the impact of the jack material (steel) on the test results, wood blocks (the axial dimension of the pipe is 20 mm, the radial dimension is 50 mm, and the circumferential dimension is 600 mm) are used to isolate the jack from the pipe. The MFL detector of the pipeline is pulled uniformly from the right side of the pipeline to the left side of the pipeline by a hoist, and the three-axis data of the MFL field of the pipe wall are collected. The internal detection experiment of the pipeline MFL under external load is shown in Figure 11.

As shown in Figure 11, the jack acts on the square center under the pipeline. According to the principle of material mechanics, the whole pipeline will produce bending stress when different loads are applied to the pipeline. The jack action area will produce a local stress concentration phenomenon. With the use of a winch to drag the in-pipe detector through the Q235 pipe at a uniform speed, the circumferential array of discharged magnetic sensors is used to collect and store the three-axis data on the leakage magnetic field of the pipe. Figure 12 and Figure 13 depict the physical object and fundamental block diagram of a pipeline MFL detection system.

The pipeline detection system developed by our team is 2.4 m in length, 100 kg in weight, 0–75 °C in running temperature, 1–12 MPa in running pressure. The probe is arranged along with the circumferential array for a week. The number of probes is 24, the number of sensors is 96, the number of three-axis channels is 384, the mileage wheel is 2, the running speed is 0.1–5 m/s, the detection rate is 1 KHz. The sensitivity of the sensor is 2.403 μT/LSB in the *x*-axis and *y*-axis, 3.872 μT/LSB in the *z*-axis.

The pipeline MFL detection system consists of four parts: a power supply section, an excitation detection section (permanent magnets with different poles are inlaid at both ends of the yoke iron and steel brushes are pasted on the periphery of the permanent magnets to excite the pipe wall evenly), a computer section (data acquisition and storage) and a mileage wheel. The sections are connected by universal joints. 

In the detection process, the pipe wall is excited by the detection system. When the pipe wall is damaged (defects, corrosion, stress, etc.), the magnetic flux leakage signal on the pipe wall surface will change. The magnetic sensor probe collects this change and stores it in the storage unit together with mileage and other information. In the later stage, through transformation and analysis of the stored data, the type, degree and location of the pipe wall damage can be judged.

#### 4.2.2. Experimental Result Analysis

To obtain the relationship between the excitation intensity and the magnetic induction intensity of the outer wall of the pipeline, the U-shaped magnet of the winding coil is fixed on the side of the steel plate (1.5 m long and 0.45 m wide) consistent with the thickness of the pipeline, and the magnetic core of U-type excitation material is consistent with the experimental material. The excitation intensity is controlled by changing the coil current. The surface magnetic induction intensity of the positive center of the U-shaped magnet on the other side of the steel plate under different excitation currents is measured by a Tesla meter. The experimental setup is shown in Figure 14.

With the use of the device in Figure 14, when the excitation current varies from 1 A to 15 A (at a 1 A progression), the relationship between the excitation current and the magnetic induction intensity on the outer wall surface can be obtained. Excitation current and magnetic field intensity experimental results are displayed in Figure 15.

Figure 15 shows that the magnetic induction intensity on the outer surface of the steel plate is basically unchanged after reaching 39 mT, i.e., the steel plate reaches magnetic saturation, and the external magnetic field strength is 20 KA/m. According to the theoretical model, when the external magnetic field is 5 KA/m, the influence of stress on the MFL field intensity of the pipeline is obvious. As a result, the magnetization is optimal when the outside surface of the steel plate reaches 9.9 mT. 

A pipeline detector is placed in the Q235 pipeline. With the excitation unit adjusted so that the magnetic induction strength of the outer wall surface is 9.9 mT, the tube wall is considered to reach the magnetization state for optimal stress detection. According to the analysis of material mechanics, when the center below the pipeline is subjected to external load in the radial direction, the external load action zone (the center below the pipeline) will produce local stress concentration. We assign the coordinate origin to the midpoint of the local stress concentration zone. Figure 16 shows the axial and radial components of the detection data of the detector in the pipeline just below the pipeline along the axial direction when the stress intensity in the local stress concentration zone reaches 150 MPa. To ensure the consistency of signal coordinates, the basic values of the axial magnetic flux leakage intensity of 5.4 mT and radial magnetic flux leakage intensity of 2 mT are given to the classical model and the improved model.

As can be seen from Figure 16, Figure 16a,b show the comparison between the classical model, the improved model, and the experimental data. Due to the noise of the probe itself, the changing trend of the experimental data is fluctuating (including random noise). In the practical application of the pipeline detection system, the FFT filtering is used for the inspection, data were processed, and better results were obtained. Therefore, Figure 16c,d show the comparison of the experimental data processed by the classical model, the improved model, and the FFT filter. As shown in Figure 16c,d, the axial component of points and two minimum points in the axial component of the MFL signal of the classical model are axisymmetrically distributed. The radial component is symmetrically distributed in the center of the local stress concentration zone, and there are peaks or troughs at the edge of the stress concentration area. The axial component distribution of the MFL signal in the improved model is slightly different from that in the classical model, which has only one minimum value and is axisymmetrically distributed about the minimum point, and the axial and radial components of the magnetic flux leakage signal change more slowly. The variation trend of the axial component and radial component of the MFL signal in the improved model is closer to the experimental results. The experimental axial data have a little deviation, which is because the pipeline has certain bending stress during the experiment.

At the same time, in practice, noise from oxidation of the pipe wall, bending stresses in the pipe as a whole due to local stresses applied to the pipe, and residual stresses in the pipe itself can cause noise and deflection of the inspection signal. The reproducibility of the method is demonstrated by the good repeatability of the results of the two experiments.

Figure 17 shows the variation of axial and radial components in the local stress concentration zone of the pipe after the FFT filtering treatment for top forces of 0, 50, 100 and 150 MPa.

As can be seen from Figure 17, during the actual detection, the leakage signal in the local stress concentration zone of the pipe varies significantly with the amplitude in the axial component and peak-to-peak value in the radial component. The trends of the axial and radial components of the leakage field are consistent with the theoretical model. With the enhancement of the external load, the change in the axial and radial components of the leakage field in the local stress concentration zone of the pipe gradually increases. The axial component amplitude and radial component peak-to-peak value of the leakage magnetic field when the external load is 0, 50, 100, and 150 MPa are shown in Table 3.

The experimental results are basically consistent with the simulated variation trend. The MFL signal fluctuates in the actual detection as a result of uneven doping inside the actual pipeline, inconsistent wall thickness of the pipeline, uneven stress of the pipe body, and uneven oxidation during long-term use. The large bending stress of the pipe caused by the external load in the experiment also led to a change in the base value of the MFL signal of the pipeline.

A comparison of Figure 6, Figure 7, and Figure 17 shows that the axial and radial component distribution curves of the MFL signal calculated by the improved analytical model of MFL signal are in better agreement with the experimentally obtained curves. Through a comparison of the simulation results of the classical analytical model of the MFL signal and the improved analytical model of the MFL signal with the experimental results, the results of the improved MFL signal analytical model are closer to the actual engineering results. Therefore, the improved analytical model of the MFL signal can be used for guidance and analysis of actual engineering data.

As can be seen from Table 1, Table 2 and Table 3, a comparison of the axial component amplitude and radial component peak-to-peak values of the classical model, the improved model, and the experimental data are shown in Figure 18.

As can be seen from Figure 18, by comparing the axial component amplitude and radial component peak-to-peak value of the classical model, the improved model, and the experimental data, the improved model is closer to the experimental value in numerical value. The axial component amplitude and radial component peak-to-peak value of the two are similar, and the minimum error reaches 5%. However, when the stress intensity is 150 MPa, the radial peak value error is large. From the perspective of simulation, this is because as the stress increases, a, α and k, etc. in the model are functions of the stress. In this paper, to simplify the calculation, they are set to constant values, which will also affect the results of the improved model.

During the detection process, speed, time, and repeatability have a significant impact on the excitation effect and detection results. According to reference [33], the domain rotation process can be divided into two stages in the relaxation time. In the first stage, the magnetic domain rapidly rotates to the initial state, and the residual stress hinders the movement. In the second stage, the magnetic domain continues to move and fluctuates under the influence of lattice and geomagnetic fields. In addition, due to the factors such as the sampling rate and magnetization time of the actual detection probe, as the detection speed increases, the excitation and detection results will be adversely affected [34,35]. According to the above literature, the magnetic flux leakage signal will decay rapidly in the relaxation time. The detection effect is better when the stress hinders the rotation of the magnetic domain obviously. Therefore, the pipeline detection system uses real-time detection. In the practical application of the pipeline detection system used in this study, the operation speed of the pipeline detection system is 1–2 m/s. After many experiments, the pipeline detection system has good detection effect and repeatability.

The signal caused by local stress effect in pipeline MFL measurement is not an obvious defect. In the actual pipeline, there are many unobvious defects, such as uneven doping. To compare the MFL signal caused by local stress effect and doping inhomogeneity, the carbon content in the local area of the pipeline is reduced by decarbonizing the local area of the pipeline. The processing area is a circle of 20 mm in diameter, forming an uneven doping region. The dragging experiment was carried out on the decarburization region. The comparison between the magnetic flux leakage signal caused by the uneven doping of the tube and the magnetic flux leakage signal induced by the local stress is shown in Figure 19.

As shown in Figure 19, the axial and radial components of MFL signal caused by the stress effect are 0.640 mT and 1.731 mT, respectively. The axial and radial components of MFL signals caused by uneven pipe doping are 0.727 mT and 0.837 mT, respectively. The two are in the same order of magnitude. However, the MFL signal caused by the stress effect is more regular than that caused by decarburization, and the peak value of magnetic flux leakage caused by stress is more prominent. Due to the influence of the decarburization boundary, the peak value of MFL signal induced by decarburization is not sharp.

In follow-up research, the eddy current, the relevant mechanical coefficient of the material, and the residual magnetic field strength of the unmagnetized region need to be further considered. At the same time, the MFL analytical models for different stress states of the pipeline need to be further analyzed, and the MFL signal in the local stress concentration zones and defects of the pipeline should be identified. These approaches can provide a more reliable theoretical and experimental basis for quantitative research on the local stress of pipelines.

## 5. Conclusions

During pipeline operation, the pipe wall is subjected to local stress action caused by internal pressure, external load, and high temperature. The classical analytical model of the MFL signal is analyzed for the magnetic charge model of pipeline crack. However, no effective research method for the MFL signal generated by continuous local stress distribution is available. Based on the Jiles–Atherton model, this paper establishes the improved analytical model of the MFL signal based on the magnetomechanical coupling-type pipeline local stress concentration zone, which provides a scientific basis for the detection and quantitative research of pipeline local stress concentration zone. This paper obtained the following conclusions:Considering the hysteresis condition, the hysteresis loop of Q235 material shows a downward trend as the stress increases.When the external magnetic field is about 5 KA/m, the stress has the most obvious influence on the magnetization of ferromagnetic materials, which is considered to achieve the best magnetization of stress detection.The MFL signal is abnormal in the local stress concentration zone of the pipeline, and the variation of axial amplitude and radial peak-to-peak value of the MFL signal increases with the increase in stress intensity. The axial amplitude and radial peak-to-peak value of the MFL field can reflect the variation of the MFL signal characteristics in the local stress concentration zone.A comparison of the simulation results with the experimental results shows that the simulation results of the improved analytical model of MFL signal are in good agreement with the experimental results. Therefore, the improved analytical model of MFL signal is more suitable for the quantitative study of the local stress concentration zone of the pipeline.

## Figures and Tables

**Figure 1 sensors-22-01128-f001:**
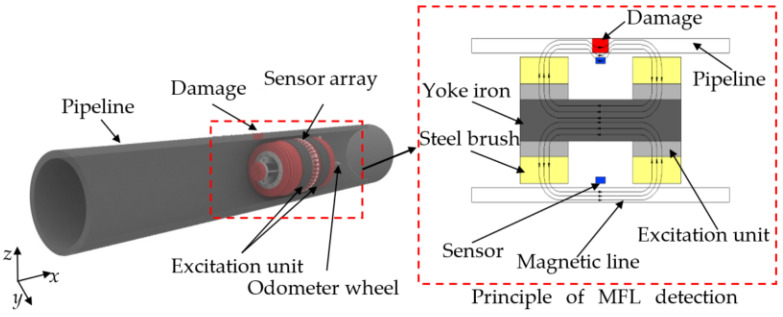
Three-dimensional schematic of pipeline and MFL detection equipment.

**Figure 2 sensors-22-01128-f002:**
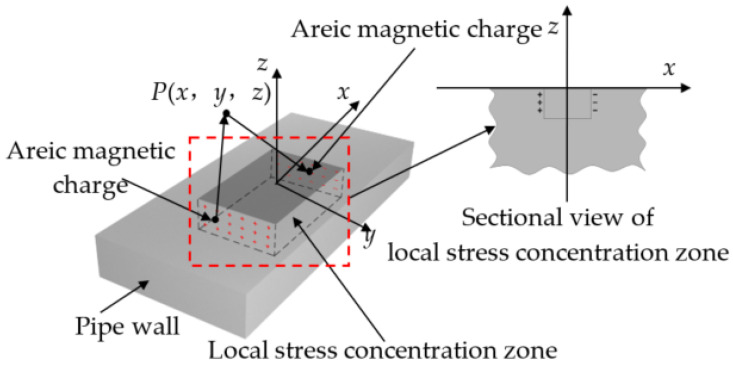
Schematic of the classical model for 3D pipeline local stress concentration zone.

**Figure 3 sensors-22-01128-f003:**
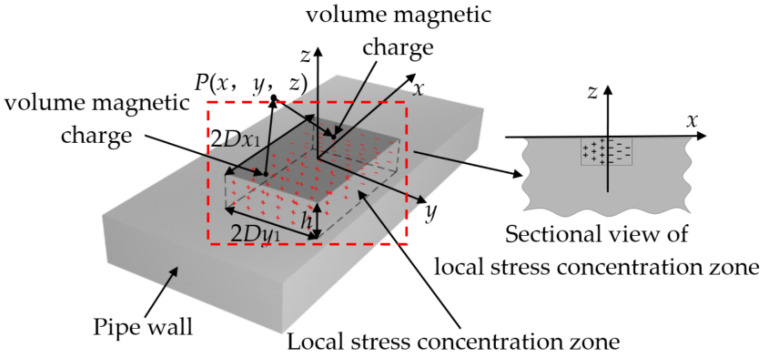
Schematic of the improved model for 3D pipeline local stress concentration zone.

**Figure 4 sensors-22-01128-f004:**
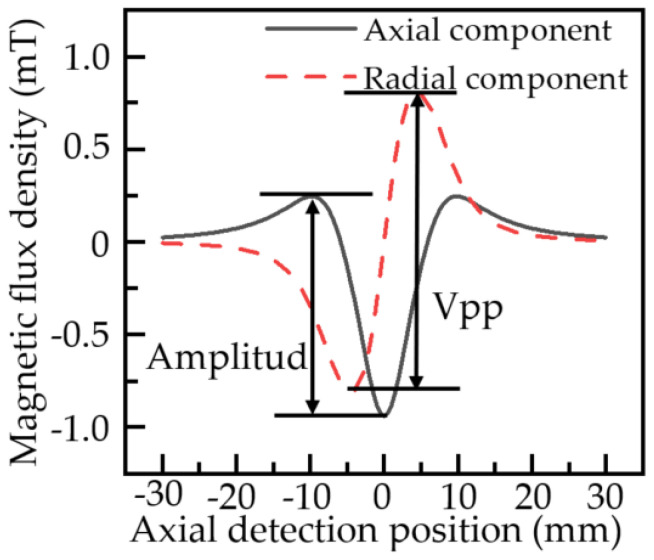
Distribution curve of MFL signals of the pipeline local stress concentration zone.

**Figure 5 sensors-22-01128-f005:**
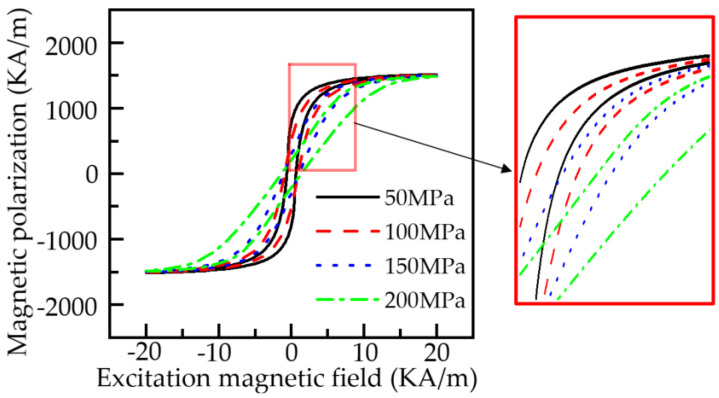
Hysteresis loops of ferromagnetic materials under different stresses.

**Figure 6 sensors-22-01128-f006:**
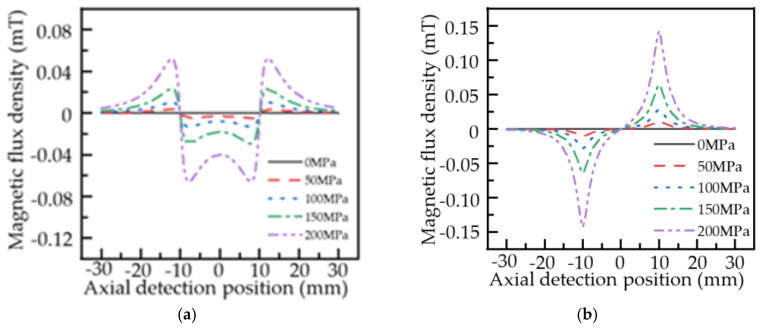
Distribution of the MFL signal of the pipeline local stress concentration zone under different stresses based on the classical model: (**a**) axial component; (**b**) radial component.

**Figure 7 sensors-22-01128-f007:**
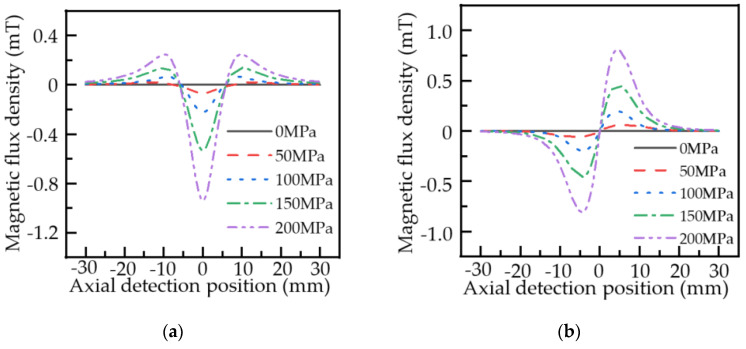
Distribution of the MFL signal of the pipeline local stress concentration zone under different stresses based on the improved model: (**a**) axial component; (**b**) radial component.

**Figure 8 sensors-22-01128-f008:**
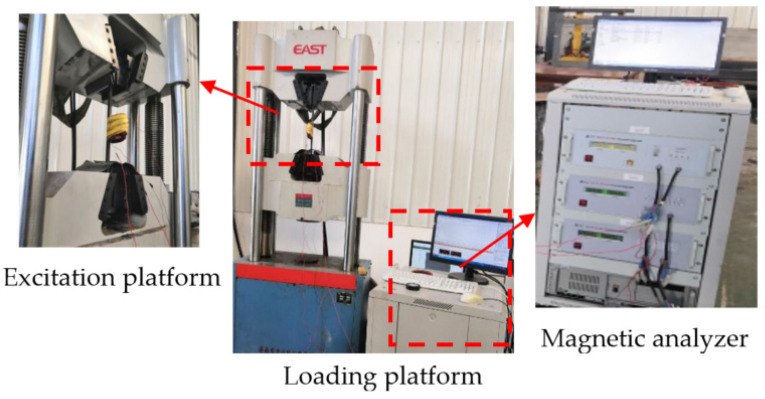
Experimental diagram of hysteresis loop test of ferromagnetic materials.

**Figure 9 sensors-22-01128-f009:**
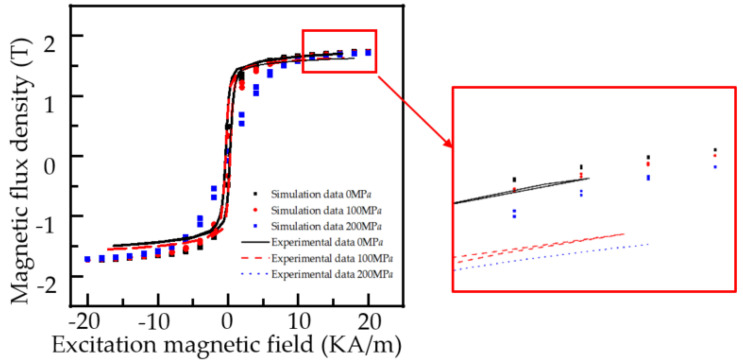
Hysteresis curve of Q235 steel under different stresses when the maximum external magnetic field is 20 KA/m.

**Figure 10 sensors-22-01128-f010:**
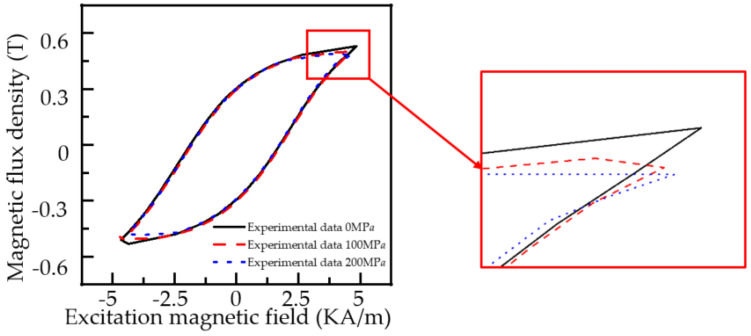
Hysteresis curve of Q235 steel under different stresses when the maximum external magnetic field is 5 KA/m.

**Figure 11 sensors-22-01128-f011:**
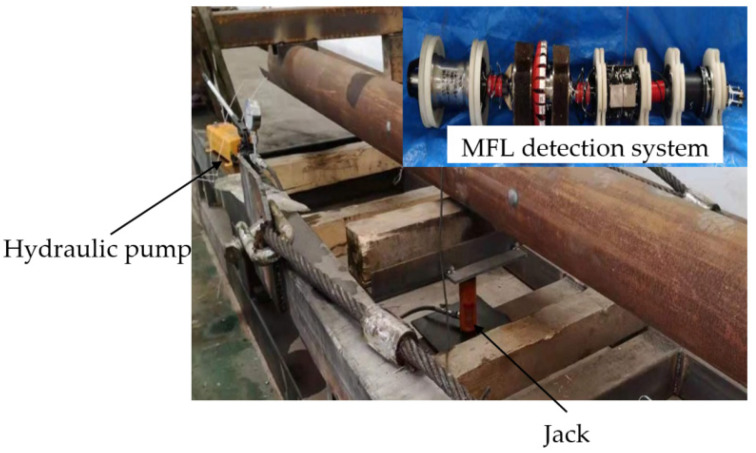
Experimental diagram of MFL signal detection of the pipeline local stress concentration zone under external load.

**Figure 12 sensors-22-01128-f012:**
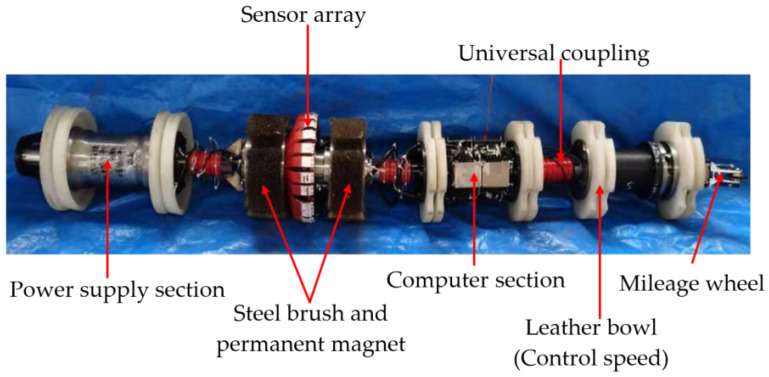
The physical representation of a pipeline MFL detection system.

**Figure 13 sensors-22-01128-f013:**
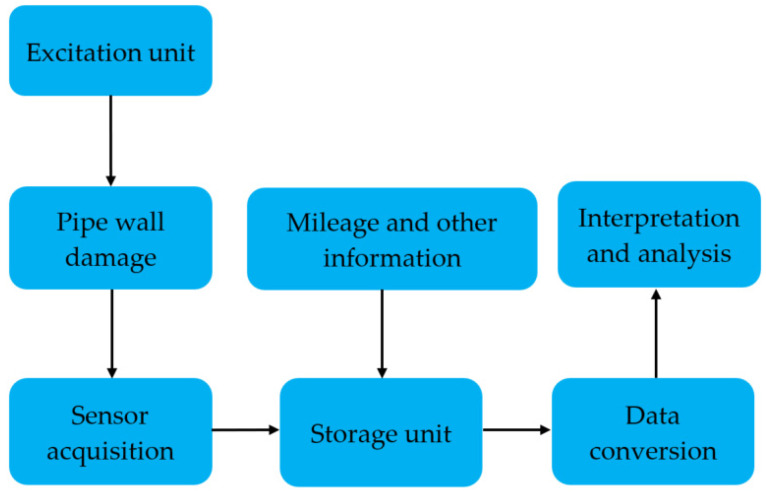
The principle block diagram of a pipeline MFL detection system.

**Figure 14 sensors-22-01128-f014:**
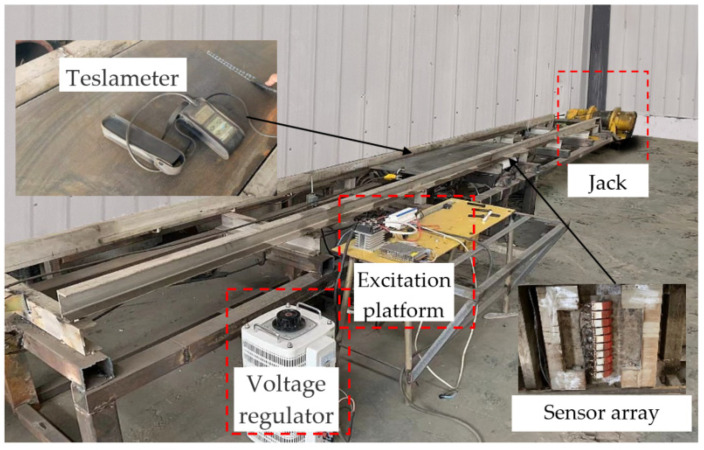
Experimental diagram of the Q235 steel plate excitation device.

**Figure 15 sensors-22-01128-f015:**
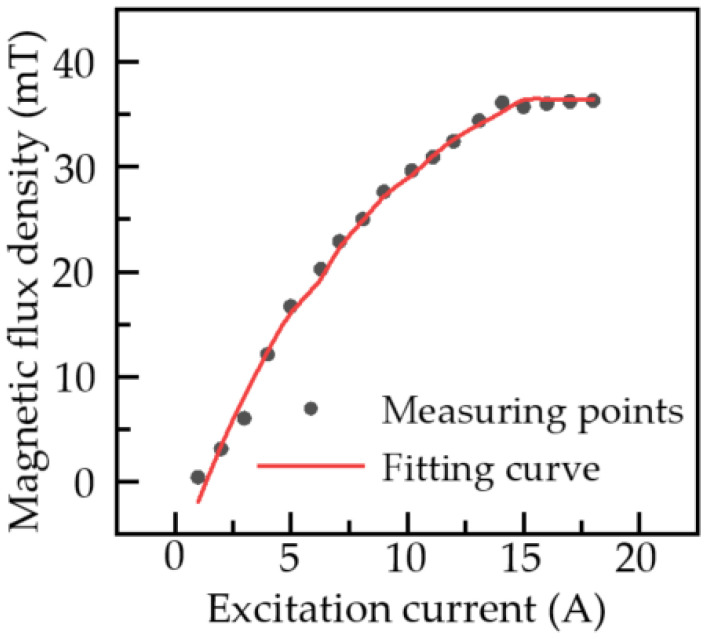
Tendency diagram of magnetic induction intensity on the outer surface of the steel plate under different exciting currents.

**Figure 16 sensors-22-01128-f016:**
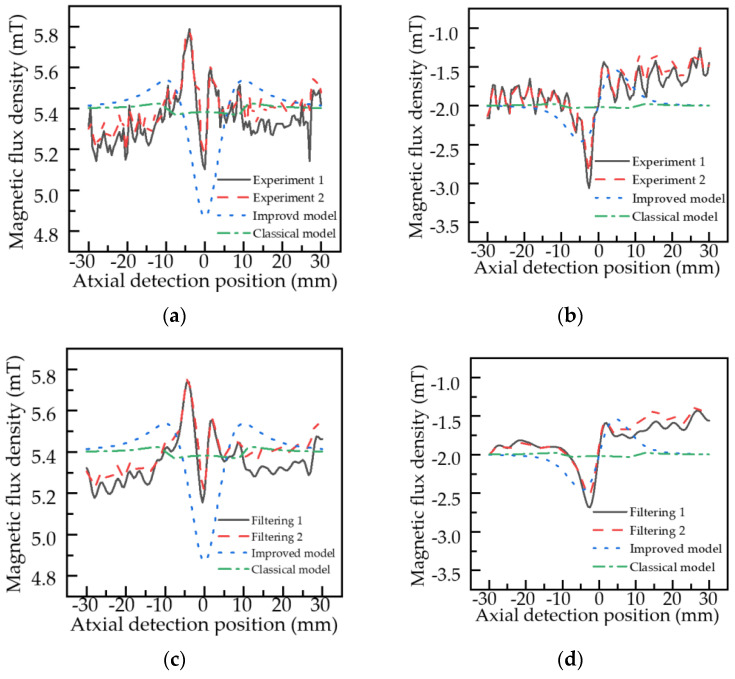
Distribution of the MFL signal of the pipeline local stress concentration zone: (**a**) axial component; (**b**) radial component; (**c**) axial component after FFT filtering; (**d**) radial component after FFT filtering.

**Figure 17 sensors-22-01128-f017:**
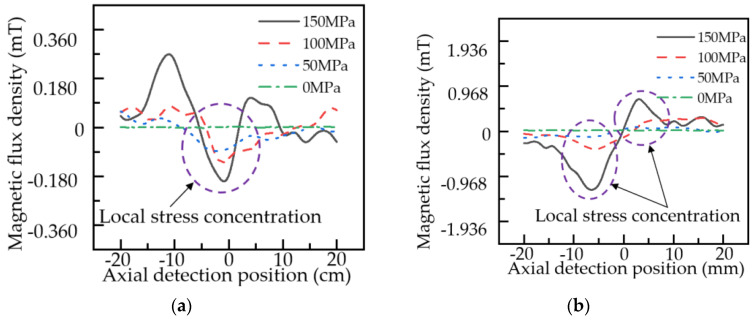
Distribution of the MFL signal of the pipeline local stress concentration zone under different stresses: (**a**) axial component; (**b**) radial component.

**Figure 18 sensors-22-01128-f018:**
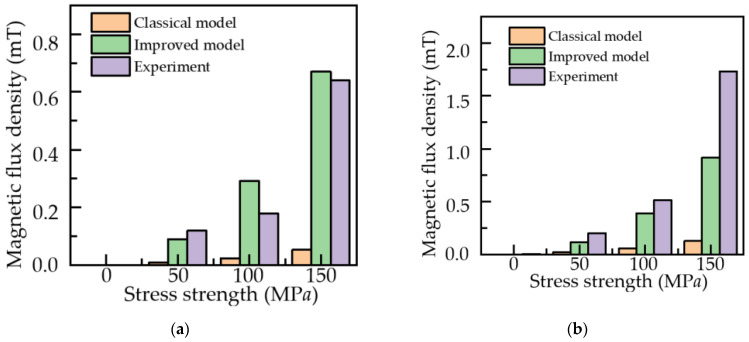
Comparison chart of classical model, improved model and experimental data: (**a**) axial component amplitude; (**b**) radial component peak-to-peak.

**Figure 19 sensors-22-01128-f019:**
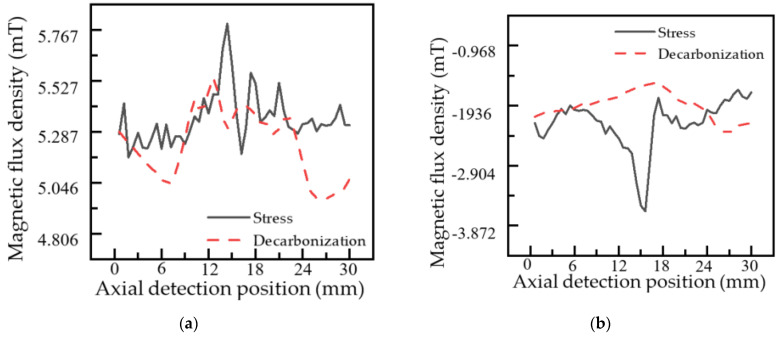
Distribution of the MFL signal of the pipeline local stress concentration zone under stress and quench: (**a**) axial component; (**b**) radial component.

**Table 1 sensors-22-01128-t001:** Variation of axial component amplitude and radial component peak-to-peak value of MFL signals based on classical model.

Stress Intensity (MPa)	Axial Amplitude (mT)	Radial Peak-to-Peak (mT)
0	0	0
50	0.009	0.022
100	0.024	0.057
150	0.053	0.128
200	0.118	0.284

**Table 2 sensors-22-01128-t002:** Variation of axial component amplitude and radial component peak-to-peak value of MFL signals based on the improved model.

Stress Intensity (MPa)	Axial Amplitude (mT)	Radial Peak-to-Peak (mT)
0	0	0
50	0.090	0.116
100	0.291	0.390
150	0.670	0.914
200	1.186	1.615

**Table 3 sensors-22-01128-t003:** Variation of axial component amplitude and radial component peak-to-peak value of MFL signals in the experiment.

Stress Intensity (MPa)	Axial Amplitude (mT)	Radial Peak-to-Peak (mT)
0	0.001	0.004
50	0.119	0.204
100	0.179	0.513
150	0.640	1.731

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
