# Peer review of "Research on the Analytical Model of Improved Magnetic Flux Leakage Signal for the Local Stress Concentration Zone of Pipelines"

_sensors, 2022, doi:10.3390/s22031128_

Round 1

Reviewer 1 Report

Lijian Yang et al study the effect of stress on the MFL signal. This study is potential for exploration in the MFL measurements. The manuscript is well constructed, I am glad to read it.

The origin of the stress effect on the MFL signal is that the stress causes an induced anisotropy of the ferromagnetic pipe (sample), the MFL measurement can detect this small change.

However, the signal caused by the stress effect in the MFL measurement is the result of a non-obvious defect. The non-obvious defects include uneven doping as well (as stated by the authors). Therefore the author should have comparative measurements to compare the orders of magnitude of MFL signals of these two effects if they coexist in the pipe.

Another important comparison that should be done is with the distorted signal coming from the Wobble effects... which is not coming from any defect.

Reviewer 2 Report

Journal Editorial Board/ Staff is responsible of the detection of the plagiarism and inappropriate self-citations by authors. The reviewer did NOT test the paper.

Please indicate the original achievements of the authors of the work.

Please indicate the advantages of the solution proposed by the authors as a result of quoting literature sources of known and used solutions in the field of examining the technical condition of pipelines in the operation process.

Reviewer 3 Report

Dear Authors,

Congratulations on interesting research.

Please kindly revise the paper accordingly with the following comments :

  • The introductory section should be amended with additional review of recent research on magnetoelastic effect in steel, measurement and moddeling of the effects of stress on hysteresis loops, mesurement ans modelling of anhysteretic curves for Jiles-Atherton model, recent corrections for Jiles-Atherton model of hysteresis loop, stress range/values exhibited by pipelines under operating conditions.
  • please state if the stresses applied in modelling and measurement of hysteresis loops were tensile or compressive, as the hysteresis loop behavior is different due to stress sign - was it accounted for in the research? Additionaly, existence of Villari point may be detrimental for one kind of stress (tensile or compressive, depending on magnetostriction sign) due to unequivocality of the results, please explain if it was tested
  • please directly compare (on one figure ) results of modelling and measurement of hysteresis loops
  • are results for hysteresis loops measured with 5kA/m are available? Showing only the 5kA/m area for saturated hysteresis loops is qualitatively same, but quantitatively different
  • please directly compare (on one figure) the results for modelling (fpr classical and improved model) and measurements of MFL signal, please explain inevitable differences
  • please provide better quality for photographs of the measurement systems , maybe make them bigger (this is interesting!). Please provide schematic block diagrams along them.
  • Please provide details of the measurement systems (equipment used etc.)
  • figure 12 - the fitting line used is non-physical nonsense, please change it for something simpler, e.g. Langevin equation or other saturating function.
  • Please comment if the saturation observed in figure 12 is due to saturation of the test material, or magnetizing yoke
  • Please provide results for two separate test runs in figure 13 to asses repeatability
  • are the results in figure 14 averaged from many test runs, or filtered? please explain.

best regards,

Reviewer

Round 2

Reviewer 1 Report

I believe that readers will have a lot of arguments with the authors in drawing such superficial conclusions on the comparison of the MFL signals of non-obvious defects and system error.

In the comparison of the two types of non-obvious defects, the stress defect can be distinguished from the non-even doping defect if the stress defect is a local defect only. It's not easy to distinguish between the large area stress defect and non-even doping case. In the large area stress defect case, the classical model could be more suitable than your improved model.

Another point, with recent improvement, the MFL signal due to wobble effect can be achieved at sub-Gauss level; this signal is in the same order as the MFL signal caused by the stress defect. However, we agree with the authors that the MFL signal waveform due to the wobble effect could be similar to the case of non-even doping.

To this end, I recommend authors to revise their manuscript for the study cases of small or local stress defects only.    
